# How Do Chinese National Scenic Areas Affect Tourism Economic Development? The Moderating Effect of Time-Limited Rectification

**DOI:** 10.3390/ijerph182111620

**Published:** 2021-11-05

**Authors:** Yongcuomu Qu, Ziqiong Zhang, Yanchao Feng, Xiaorong Cui

**Affiliations:** 1School of Economics and Management, Harbin Institute of Technology, Harbin 150001, China; qycm@hit.edu.cn (Y.Q.); ziqiong@hit.edu.cn (Z.Z.); 2Business School, Zhengzhou University, Zhengzhou 450001, China; 3College of Physics and Electronic Information, Inner Mongolia Normal University, Hohhot 010022, China; 20194016010@mails.imnu.edu.cn

**Keywords:** Chinese national scenic areas, tourism economy, moderating effect, time-limited rectification

## Abstract

Based on panel data on 124 prefecture-level and above cities from 2003 to 2018, this study investigated the impact of CNSAs on tourism economic development and the moderating effect of time-limited rectification by comprehensively using the quasi-DID model, the static spatial Durbin model, and the dynamic spatial Durbin model. The results showed that the impact of CNSAs on tourism economic development has a heterogeneous characteristic in terms of tourists and revenue. In addition, the spatial spillover effect and the path dependence have effectively promoted tourism economic development. Furthermore, the effectiveness of time-limited rectification has been proved in this study, while the “beggar-thy-neighbor” effect has, to some extent, weakened the promotional effect of CNSAs on tourism economic development, especially in terms of international tourists and international tourism revenue. Finally, relevant policy implications for the superior department in charge, local governments, and the management department of CNSAs are outlined to provide a practical reference for promoting the high-quality development of the tourism economy in China.

## 1. Introduction

Faced with the dichotomous challenge of balancing the quest for economic growth and its associated risks for the environment, the tourism economy has taken central stage among scholars and policymakers at home and abroad for its features of cleaner production and sustainability [1,2,3,4]. In particular, over the past 40 years, China’s tourism industry has stimulated economic activities thereby serving as a great catalyst for economic growth [5]. Furthermore, to achieve the dual goals of protection and development, the Chinese authorities have issued various policies to promote the development of the regional tourism economy based on protecting natural and cultural resources [5,6].

As important support of tourism development, Chinese national scenic areas (hereafter CNSAs) have been acknowledged as an important engine of regional economic growth by driving infrastructure construction, increasing employment opportunities, and promoting industrial structure upgrading [6]. Since the first list of 44 CNSAs was issued in 1982, a total number of nine lists including 244 CNSA have been acknowledged as of 2018. In particular, to show China’s major scenic areas conveniently, competent authorities have set CNSAs as top-level scenic spots [7,8].

Generally, the application of a CNSA is submitted by the provincial-, autonomous regional-, and municipal-level government and jointly evaluated by the relevant construction department under the State Council, the relevant Environmental Protection Department under the State Council, the relevant forestry department, and the relevant cultural relics department, and finally approved and announced by the State Council [9]. The clear definition of a CNSA using the assessment of landscape resources, measurement for ecological resources protection, layout of major construction projects, intensity of development and utilization, functional structure, and spatial layout prohibit or restrict development areas, and tourist capacity is necessary for achieving the optimal balance of tourism-focused economic development and conservation of environmental and cultural resources [10,11].

However, CNSAs have been regarded as permanent and unregulated for a long time, and this situation inevitably leads to complex situations of weak impetus, poor planning, destruction of resources, and unreasonable exploitation [7,12,13]. At the same time, the relevant authorities have lacked the corresponding assessment, resulting in the delayed construction and poor service of CNSAs which, in turn, erodes their “brand effect” [14,15]. Moreover, due to the huge benefits that CNSAs bring to local governments, they tend to follow up on applications but fail to follow through after authentication [16].

Against this background, the Ministry of Housing and Urban–Rural Development launched a four-year program to rectify the aforementioned problems during the period of 2012–2015, ordered problematic CNSAs to be rectified within one year, and reviewed the final effect of time-limited rectification in 2016. In particular, the main contents of the program included five aspects: institutional construction, planning management, construction management, service management, and image publicity. In addition, the evaluation results were divided into five grades: excellent, good, qualified, need to be rectified, and unqualified. When categorized as needs to be rectified or unqualified, the CNSA was ordered to make time-limited rectification within one year. If the substandard CNSA still failed to satisfy the standard after time-limited rectification, it would be listed as disqualified by the State Council.

According to the inspection results of the Ministry of Housing and Urban–Rural Development, there were 70 cases (times) where CNSAs were warned and put on the list of needing to be rectified, and there were 29 cases (times) where they failed to meet the standards and were placed on the list of unqualified. Fortunately, from the time-limited rectification results announced in 2016, 42 CNSAs have passed the inspection, while seven CNSAs still failed to pass. In particular, the prominent problems of increasing prices casually, ticket within ticket, compulsory consumption behavior, and natural resource destruction resulted in complaints from tourists, weakened the “advertising effect” of CNSAs, and, thus, need to be rectified.

Nevertheless, as for the moderating effect of time-limited rectification on the nexus between CNSAs and tourism economic development, existing research has not been fully explored, which leaves an incentive and opportunity for this study. In particular, whether or not CNSAs have a spatial spillover effect on tourism economic development, aside from local effects, has yet to be comprehensively studied. In addition, the effects of CNSAs on local tourism economic development before and after time-limited rectification has yet to be explored. Therefore, the aim of this study was to explore the effects of CNSA on tourism economic development and the moderating effect of time-limited rectification by comprehensively and systematically employing a series of econometric methods.

The marginal contributions of this study can be drawn from two aspects. Theoretically, to the best of our knowledge, this study analyzed the effect of CNSAs on tourism economic development for the first time and illustrates the theoretical basis for how time-limited rectification affects tourism economic development at the macroeconomic level. Methodologically, we treated time-limited rectification as a quasi-natural experiment and evaluated its effect on tourism economic development by using panel data on prefecture-level and above cities. More importantly, this study provides a normative theoretical explanation and systematic empirical facts, which aim to drive the sustainable development of tourism economy and provides decision-making guidance for strengthening the reputation of CNSAs.

The remainder of this study proceeds as follows: Section 2 provides a literature review. Section 3 presents the methodology including variables selection, data sources, and models specification. Section 4 reports and analyzes the empirical results based on three different econometric models, followed by an in-depth discussion in Section 5. Concluding remarks, policy implications, and research prospects are provided in Section 6. To vividly illustrate the research steps in this study, we drew them in Figure 1.

## 2. Literature Review

Noted as the world’s largest service sector, tourism plays an important role in boosting the sustainable development of the entire economy, while disorganized tourism development has aggravated the conflict between tourism economic development and tourism resources protection [17,18]. Against the background of an ecological civilization construction strategy in the new era, plenty of studies have focused on exploring how to realize tourism economic development without destruction of tourism resources [19,20], how to improve the service quality of the tourism industry [21,22], and how to promote the sustainable, healthy, and high-quality development of the tourism economy [23,24].

Until now, both domestic and international scholars have explored the effects of tourism on regional economic development [25,26], the relationship between tourism economic development and ecological environment [27,28], and the influencing factors of tourism efficiency [29,30], which provide ample references for this study. Specifically, although some studies have gradually paid attention to investigate the economic and environmental effects of CNSAs in China [31], there is a paucity of studies investigating how CSNAs affect tourism economic development from the dual perspectives of tourists (i.e., the number of international tourists and the number of domestic tourists) and revenue (i.e., international tourism revenue and domestic tourism revenue), notwithstanding how time-limited rectification moderates the nexus between CNSAs and tourism economic development.

As for the econometric model of policy evaluation, most of the existing literature employed the difference-in-differences (DID) model, which should pass the parallel trend test [32], the placebo test [33], and the counterfactual test [34]. However, the core explanatory variable, that is, the number of CNSAs, is a continuous index and not a binary index; thus, treating the announcement of CNSAs as an unbalanced quasi-experiment, this study aimed to employ the quasi-DID model as the basic model [35]. More importantly, the spatial spillover effect was usually neglected across regions in the empirical analysis of the tourism economy, which may reduce the robustness of the estimation results; thus, it is necessary and important to adopt the spatial econometric model [36,37]. Furthermore, except for the spatial spillover effect, ignoring temporal inertia may also lead to endogenous problems; therefore, the adoption of a dynamic spatial econometric model was also necessary for comparison to ensure the robustness of the empirical results [38].

Therefore, to fill the above-mentioned research gaps and promote the high-quality development of CNSAs simultaneously, this article attempted to explore the impact of CNSAs on tourism economic development and the moderating effect of time-limited rectification by comprehensively using the quasi-DID model, the static spatial Durbin model, and the dynamic spatial Durbin model, which can, to some extent, guarantee the robustness and reliability of the empirical results.

## 3. Methodology

### 3.1. Variables Selection

Dependent variables: four indicators were employed to act as the proxy variables of tourism economic development such as the number of international tourists (*y*_1_), the number of domestic tourists (*y*_2_), international tourism revenue (*y*_3_), and domestic tourism revenue (*y*_4_).

Core explanatory variables: The number of CNSAs (*d*_1_) was employed to act as the benchmark independent variable. In addition, a dummy indicator (*d*_2_) was employed to act as the proxy variable of time-limited rectification, which equaled −1 when a CNSA belonged to the list of not up to standard and 1 otherwise.

Control variables: To capture the character of the city where a CNSA was located, several control variables, such as economic development (*x*_1_, measured by per capita gross domestic product), consumption (*x*_2_, measured by per capita retail sales of consumer goods), industrial structure upgrading (*x*_3_, measured by the shares of tertiary industry in the local gross domestic product), infrastructure (*x*_4_, measured by per capita urban road area), green rate (*x*_5_, measured by the green coverage rate of built-up area), the air pollution of sulfur dioxide (*x*_6_, measured by per capita emission of sulfur dioxide), the air pollution of dust (*x*_7_, measured by per capita emission of dust), and the liquid pollution of wastewater (*x*_8_, measured by per capita emission of wastewater), were employed by referring to the studies in [4,7,12].

### 3.2. Data Sources

The samples used in this study consisted of 124 prefecture-level and above cities that had at least one CNSA in 2018, and the investigation period covered from 2003 to 2018. Several data sources were used, which were the China City Statistical Yearbook (2004–2019), the China Tourism Yearbook (2004–2019), and the website of the Ministry of Housing and Urban–Rural Development in China (www.mohurd.gov.cn, accessed on 3 March 2021). In particular, the original data for the control variables were collected from the China City Statistical Yearbook (2004–2019), the original data for the four dependent variables were collected from the China Tourism Yearbook (2004–2019), while the time-limited rectification information on CNSAs was collected from the website of the Ministry of Housing and Urban–Rural Development in China (www.mohurd.gov.cn, accessed on 12 May 2021). To eliminate the potential interference of heteroscedasticity, all the dependent and control variables incorporated into the regression equation were in the natural logarithm form, and there statistical description are shown in Table 1.

### 3.3. Models Specification

To investigate the effects of CNSA on tourism economic development and the moderating effect of time-limited rectification comprehensively, this study employed three econometric methods including the quasi-DID model, the spatial Durbin model, and the dynamic spatial Durbin model.

Following the study of Yang et al. [35], we first used the quasi-DID method to investigate the impact of CNSAs on tourism economic development. The difference between this method and the standard DID method is that we used a continuous treatment (i.e., the number of CNSAs) to capture the relative impact. In particular, the corresponding econometric model was established as follows:(1)$$y_{it}=\alpha_{0}+\alpha_{1}d_{1}+\beta x_{it}+f_{i}+f_{t}+prov_{j}*year_{t}+\varepsilon_{it}$$
where *i* and *t* denote the city and the time, respectively; *y* denotes the dependent variables (i.e., *y*_1_, *y*_2_, *y*_3_ and *y*_4_), *α*_0_ denotes the constant term; *d*_1_ denotes the number of CNSAs; *α*_1_ denotes the coefficient of *d*_1_; *x_it_* denotes a vector of control variables; *β* denotes the coefficients of *x_jt_*; *f_i_* denotes the city fixed effect; *f_t_* denotes the time fixed effect; *prov_j_* ∗ *year_t_* denotes the joint fixed effect of province and time to capture the impact of provincial tourism policy; *ε_it_* denotes the error term.

To investigate the moderating effect of time-limited rectification on the nexus between CNSAs and tourism economic development, the interactive term *d*_1_ ∗ *d*_2_ was incorporated into the equation after being centralized.
(2)$$y_{it}=\alpha_{0}+\alpha_{2}d_{1}*d_{2}+\beta x_{it}+f_{i}+f_{t}+prov_{j}*year_{t}+\varepsilon_{it}$$
where *d*_2_ denotes a dummy variable that is equal to −1 when a CNSA is listed in the time-limited rectification, and 1 otherwise, while the other parameters are consistent with Equation (1).

Since Equations (1) and (2) belong to the nonspatial model, it implicitly assumes that there is no spatial spillover effect. However, the changes in CNSAs in local cities usually affect tourism economic development in surrounding cities, especially in the long term. In addition, tourism economic development usually has the characteristic of spatial correlation. Hence, it was necessary to take into account such a spatial lagged effect to obtain an accurate result. Accordingly, we added the spatial lag term of the dependent variables and the independent variables including the core explanatory variables and the control variables into Equations (1) and (2) to investigate whether there were spatial spillover effects from them. Thus, referring to the study of Elhorst [36], the static spatial Durbin model was employed to carry out the estimation as follows:(3)$$y_{it}=\rho W*y_{it}+\alpha_{1}d_{1}+\beta_{j}x_{jt}+\theta_{1}W*d_{1}+\varphi_{j}W*x_{jt}+f_{i}+f_{t}+\varepsilon_{it}$$
(4)$$y_{it}=\rho W*y_{it}+\alpha_{2}d_{1}*d_{2}+\beta_{j}x_{jt}+\theta_{2}W*d_{1}*d_{2}+\varphi_{j}W*x_{jt}+f_{i}+f_{t}+\varepsilon_{it}$$
where *W* denotes the spatial weight matrix [37]; *ρ* denotes the spatial coefficient of the dependent variable; *θ*_1_, *θ*_2_, and *φ_j_* denote the spatial lag coefficients to be estimated; the other parameters are consistent with Equation (2).

In addition, considering the possible path dependence of tourism economic development and the possibility of endogenous causality between CNSAs and time-limited rectification and other factors, the lag phase of the dependent variable was introduced into the static spatial Durbin model to formulate the dynamic spatial Durbin model [38].
(5)$$y_{it}=\tau y_{i,t-1}+\rho W*y_{it}+\alpha_{1}d_{1}+\beta_{j}x_{jt}+\theta_{1}W*d_{1}+\varphi_{j}W*x_{jt}+f_{i}+f_{t}+\varepsilon_{it}$$
(6)$$y_{it}=\tau y_{i,t-1}+\rho W*y_{it}+\alpha_{2}d_{1}*d_{2}+\beta_{j}x_{jt}+\theta_{2}W*d_{1}*d_{2}+\varphi_{j}W*x_{jt}+f_{i}+f_{t}+\varepsilon_{it}$$
where *y*_*i*,*t*−1_ denotes the dependent variables of the first lag phase used to control and examine the time lag effect of their changes; *τ* denotes the temporal lag coefficient of the dependent variables; the other parameters are consistent with Equation (4).

## 4. Empirical Results and Analysis

### 4.1. Nonspatial Results and Analysis

The estimation results of Equations (1) and (2) are shown in Table 2. In Columns (1)–(4), we examined the impacts of CNSAs on tourism economic development; in Columns (5)–(8), we replaced the number of CNSAs (i.e., *d*_1_) with its interactive term and time-limited rectification (i.e., *d*_1_ ∗ *d*_2_) and re-estimated the impacts. It can be noted that the coefficients of *d*_1_ in Columns (1) and (3) were significantly positive, while its coefficients in Columns (2) and (4) were positive but insignificant; in other words, without the consideration of time-limited rectification, CNSAs have effectively attracted international tourists and increased international tourism revenue, while the corresponding impacts on attracting domestic tourists and increasing domestic tourism revenue were relatively poor. However, all the coefficients of *d*_1_ ∗ *d*_2_ in Columns (5)–(8) were significantly positive; in other words, after the implementation of time-limited rectification, CNSAs have not only effectively attracted domestic tourists and international tourists but also increased domestic tourism revenue and international tourism revenue, that is, time-limited rectification released the vitality of domestic tourism in the nonspatial analysis.

### 4.2. Static Spatial Results and Analysis

The estimation results of Equations (3) and (4) are shown in Table 3. In Columns (1)–(4), we examined the impacts of CNSAs on tourism economic development; In Columns (5)–(8), we replaced the number of CNSAs (i.e., *d*_1_) with its interactive term and time-limited rectification (i.e., *d*_1_ ∗ *d*_2_) and re-estimated the impacts. It can be noted that all spatial lag coefficients (i.e., *ρ*) were significantly positive, implying that the positive spatial spillover effect of tourism economic development in China was fully established under the formula of the spatial Durbin model. In addition, the coefficients of *d*_1_ were significantly positive in Columns (1)–(4), the spatial lag coefficients of it were significantly positive in Columns (1)–(3) but insignificantly positive in Column (4); in other words, without the consideration of time-limited rectification, CNSAs not only effectively attracted international and domestic tourists in local and surrounding cities but also increased foreign tourism revenue and domestic tourism revenue in the local and surrounding cities, while the impact of it on domestic tourism revenue in surrounding cities was relatively poor. Moreover, the coefficients of *d*_1_ ∗ *d*_2_ were significantly positive in Columns (5)–(7) but insignificantly positive in Column (8), while the spatial lag coefficients of it were insignificantly positive in Columns (5) and (7), insignificantly negative in Column (6), and significantly negative in Column (8); in other words, after the implementation of time-limited rectification, the direct and positive impacts of CNSAs on tourism economic development were retained in local cities, while the indirect and positive impacts of it on tourism economic development were no longer supported in surrounding cities, that is, the effectiveness of time-limited rectification was not supported in the static spatial analysis.

### 4.3. Dynamic Spatial Results and Analysis

The estimation results of Equations (5) and (6) are shown in Table 4. In Columns (1)–(4), we examined the impacts of CNSAs on tourism economic development; in Columns (5)–(8), we replaced the number of CNSAs (i.e., *d*_1_) with its interactive term and time-limited rectification (i.e., *d*_1_ ∗ *d*_2_) and re-estimated the impacts. It can be noted that all the spatial lag coefficients (i.e., *ρ*) were significantly positive, implying that the positive spatial spillover effect of tourism economic development in China was also fully established under the dynamic spatial Durbin model. In addition, all the temporal lag coefficients (i.e., *τ*) were significantly positive and greater than the corresponding spatial lag coefficients (i.e., *ρ*), implying that the positive temporal effect of tourism economic development in China was not only fully established under the dynamic spatial Durbin model but also had a greater impact than the spatial spillover effect of it.

In addition, the coefficients of *d*_1_ were significantly positive in Columns (1) and (3), and insignificantly positive in Columns (2) and (4), while the spatial lag coefficients were significantly negative in Columns (1) and (3) but insignificantly positive in Columns (2) and (4); in other words, without the consideration of time-limited rectification, CNSAs have merely attracted international tourists and increased international tourism revenue in local cities, while the insignificant results for domestic tourists and domestic tourism revenue in local and surrounding cities may give evidence of domestic tourists’ little interest for the homogeneous service of CNSAs. Moreover, the coefficients of *d*_1_ ∗ *d*_2_ were insignificantly positive in Columns (5)–(8), while the spatial lag coefficients of it were significantly negative in Columns (5) and (7), insignificantly negative in Column (6), and insignificantly positive in Column (8); in other words, after the implementation of time-limited rectification, the positive impacts of CNSAs on international tourists and international tourism revenue in local cities were weakened. One possible reason could be that the destination selection of international tourists has the characteristic of a “herd effect”, and they are more sensitive to negative information regarding time-limited rectification.

## 5. Further Discussion

From the estimation results in Table 2, Table 3 and Table 4, it can be drawn that the results under the dynamic spatial Durbin model, which considers both the spatial and temporal lag effects of independent variables simultaneously, had the best theoretical explanation and practical significance. Therefore, in the following discussion, we focus on the results in Table 4.

After the implementation of time-limited rectification, the services of all CNSAs ought to be improved, while the psychological shadow of international tourists pushes them to choose the tourism destinations which are not listed in the roster of time-limited rectification [39]. Thus, it is not hard to learn why the direct coefficients of *d*_1_ ∗ *d*_2_ become insignificant in Columns (5) and (7).

In addition, no matter with or without the consideration of time-limited rectification, the significant negative effects of CNSAs (or the interactive term of CNSAs and time-limited rectification) on tourism economic development were merely supported for international tourists and international tourism revenue; one possible reason may be that compared with domestic tourists, international tourists face the disadvantage of information acquisition, that is, the information asymmetry has restricted their options [40].

Generally, international tourists tend to choose CNSAs as their tourism destination, while their available time for travel is limited, which creates the problem of limited options and causes the “beggar-thy-neighbor” effect of CNSAs, that is, a vicious strategy devoted to attracting more international tourists in one city at the expense of a reduction in surrounding regions [41]. Thus, to eliminate the prevalence of the “beggar-thy-neighbor” effect among CNSAs and win a greater reputation in the world, an in-depth reform of the cooperation mechanism in the Chinese tourism industry is necessary for the future [42].

## 6. Conclusions, Policy Implications, and Research Prospects

### 6.1. Conclusions

Based on panel data on 124 prefecture-level and above cities from 2003 to 2018, this study investigated the impacts of CNSAs on tourism economic development and the moderating effect of time-limited rectification by comprehensively using the quasi-DID model, the static spatial Durbin model, and the dynamic spatial Durbin model. The main conclusions can be drawn as below.

Firstly, the impacts of CNSAs on tourism economic development can be attributed to discrete categories in terms of domestic tourists and international tourists and domestic tourism revenue and international tourism revenue. For instance, when comparing domestic versus international tourists, the information asymmetry not only increased the difficulty for international tourists in the selection of a tourism destination but also caused irrational competition among CNSAs, which formed a vicious circle.

Secondly, both the spatial spillover effect and path dependence were the impetus in promoting tourism economic development, which not only highlights the advantage of the dynamic spatial Durbin model compared with the nonspatial and static spatial econometric models but also provides evidence of cherishing the “brand effect” and “advertising effect” of CNSAs so as to achieve their sustainable development in the long run.

Thirdly, the effectiveness of time-limited rectification has been conditionally and partly proved in this study. The existence of the “beggar-thy-neighbor” effect among different cities has, to some extent, weakened the promotion effect of CNSAs on tourism economic development, especially in terms of international tourists and international tourism revenue. The realization of a more effective system in place of a vicious cycle of CNSA remains incomplete.

### 6.2. Policy Implications

The above-mentioned findings can draw the following policy implications for the related departments.

Firstly, for the superior department in charge, a CNSA is a good choice to promote tourism economic development where the incentive and restraint mechanism is necessary for its high-quality development. In particular, to take advantage of the path dependence, the evaluation criteria and supervision mechanism of CNSAs should be dynamic and normalized.

Secondly, for local governments, CNSAs can improve the reputation of regional tourism destinations in the short term, but it could lead to the problem of inaction or slackness. To win their reputation and attract more tourists in the long term, the service level of CNSAs should be promoted actively rather than rectified passively, and a periodic inspection system should be established.

Thirdly, for the management department of CNSAs, it is important and necessary to clarify the responsibility of each section rather than multi-sector coordinated management, which could reduce the transaction cost and the rent-seeking space. It is also important and necessary to avoid the vicious competition among stakeholders and promote tourism economic development with the aid of the spatial spillover effect.

### 6.3. Research Prospects

This study initially examined the moderating effect of time-limited rectification on the nexus between CNSAs and tourism economic development, while several limitations should be identified to highlight research prospects. For instance, except for CNSAs and time-limited rectification, other influencing factors, including government cooperation and environmental regulation, may have remarkable effects on tourism economic development [8,10]. In addition, to achieve an optimal balance of economic development and conservation of environmental resources, how the tourism economy affects the local ecological environment, energy consumption, and urbanization may be potential investigation directions, which also has great theoretical and practical significance for similar emerging countries [1,4].

## Figures and Tables

**Figure 1 ijerph-18-11620-f001:**
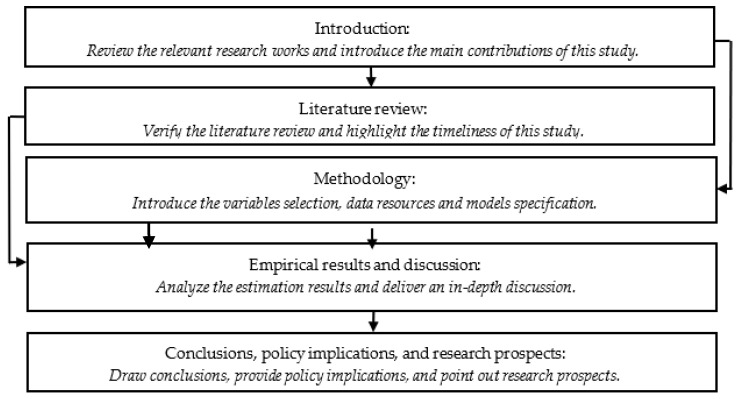
Research steps.

**Table 1 ijerph-18-11620-t001:** The statistical description of relevant variables.

Variables	Meaning	Unit	Count	Mean	SD	Minimum	Maximum
*y* _1_	The number of international tourists	Person	1984	4.637	2.033	−4.711	9.408
*y* _2_	The number of domestic tourists	Person	1984	9.665	1.171	1.099	12.561
*y* _3_	International tourism revenue	10^6^ USD	1984	3.713	2.339	−6.238	8.777
*y* _4_	Domestic tourism revenue	10^6^ CNY	1984	9.435	1.398	2.890	12.799
*d* _1_	The number of CNSAs	Piece	1984	1.225	0.705	0.000	4.000
*d* _2_	Time-limited rectification	-	1984	0.930	0.367	−1.000	1.000
*x* _1_	Economic development	CNY	1984	10.332	0.671	7.906	12.778
*x* _2_	Consumption	CNY	1984	9.535	0.823	6.011	11.592
*x* _3_	Industrial structure upgrading	%	1984	3.689	0.227	2.484	4.447
*x* _4_	infrastructure	m^2^	1984	2.180	0.630	−0.525	4.159
*x* _5_	Green rate	%	1984	3.602	0.320	−0.511	4.541
*x* _6_	The air pollution of sulfur dioxide	Ton	1984	3.851	1.076	−2.115	7.346
*x* _7_	The air pollution of dust	Ton	1984	5.714	1.243	−3.378	8.255
*x* _8_	The liquid pollution of wastewater	Ton	1984	5.014	1.277	−0.779	11.069

**Table 2 ijerph-18-11620-t002:** Results of the nonspatial econometric estimation.

Variables	*y* _1_	*y* _2_	*y* _3_	*y* _4_	*y* _1_	*y* _2_	*y* _3_	*y* _4_
	(1)	(2)	(3)	(4)	(5)	(6)	(7)	(8)
*d* _1_	0.313 **	0.084	0.304 **	0.070				
	(2.392)	(1.344)	(2.213)	(1.515)				
*d*_1_ × *d*_2_					0.053 **	0.024 **	0.069 **	0.021 **
					(1.987)	(2.074)	(2.368)	(2.466)
*x* _1_	−0.010	0.017	0.022	0.022	−0.014	0.016	0.019	0.021
	(−0.130)	(0.356)	(0.256)	(0.624)	(−0.171)	(0.343)	(0.215)	(0.608)
*x* _2_	0.048	0.054	0.143	0.019	0.035	0.051	0.131	0.017
	(0.459)	(0.988)	(1.292)	(0.564)	(0.335)	(0.938)	(1.215)	(0.481)
*x* _3_	−0.419	0.240	−0.534 *	−0.211	−0.401	0.246	−0.514 *	−0.206
	(−1.633)	(1.453)	(−1.871)	(−1.479)	(−1.568)	(1.502)	(−1.828)	(−1.427)
*x* _4_	−0.004	−0.038	0.019	−0.017	−0.014	−0.040	0.011	−0.019
	(−0.050)	(−0.790)	(0.227)	(−0.634)	(−0.181)	(−0.830)	(0.127)	(−0.694)
*x* _5_	−0.129	0.018	0.186	0.040	−0.108	0.023	0.205	0.045
	(−1.221)	(0.498)	(1.390)	(1.042)	(−0.989)	(0.647)	(1.526)	(1.157)
*x* _6_	0.016	−0.017	0.014	0.015	0.020	−0.016	0.017	0.015
	(0.374)	(−0.472)	(0.254)	(0.753)	(0.465)	(−0.460)	(0.306)	(0.751)
*x* _7_	0.017	0.090^*^	0.017	0.137 ***	0.020	0.091 *	0.019	0.137 ***
	(0.309)	(1.878)	(0.258)	(3.318)	(0.347)	(1.893)	(0.285)	(3.237)
*x* _8_	0.063	0.005	0.079	0.033 *	0.055	0.003	0.072	0.032
	(1.655)	(0.166)	(1.624)	(1.677)	(1.442)	(0.101)	(1.470)	(1.622)
constant	5.426 ***	7.546 ***	2.433	8.614 ***	5.806 ***	7.624 ***	2.760	8.677 ***
	−0.010	0.017	0.022	0.022	(3.489)	(7.239)	(1.593)	(13.017)
N	1936	1936	1936	1936	1936	1936	1936	1936
R^2^	0.954	0.940	0.955	0.980	0.953	0.940	0.955	0.980

Note: *t* statistics are in parentheses; * *p* < 0.1, ** *p* < 0.05, and *** *p* < 0.01.

**Table 3 ijerph-18-11620-t003:** Results of the static spatial econometric estimation.

Variables	*y* _1_	*y* _2_	*y* _3_	*y* _4_	*y* _1_	*y* _2_	*y* _3_	*y* _4_
	(1)	(2)	(3)	(4)	(5)	(6)	(7)	(8)
*ρ*	0.274 ***	0.395 ***	0.211 ***	0.258 ***	0.281 ***	0.399 ***	0.217 ***	0.259 ***
	(12.190)	(21.588)	(9.224)	(12.249)	(12.588)	(21.882)	(9.509)	(12.265)
*d* _1_	0.452 ***	0.129 ***	0.435 ***	0.047 **				
	(9.310)	(4.941)	(8.105)	(2.076)				
*d*_1_ × *d*_2_					0.076 ***	0.023 **	0.087 ***	0.009
					(3.750)	(2.157)	(3.890)	(1.003)
*W* × *d*_1_	0.163 **	0.079 **	0.243 ***	0.032				
	(2.305)	(2.094)	(3.119)	(0.990)				
*W* × *d*_1_ × *d*_2_					0.044	−0.008	0.040	−0.025 *
					(1.510)	(−0.534)	(1.254)	(−1.905)
Control Variables	Yes	Yes	Yes	Yes	Yes	Yes	Yes	Yes
N	1984	1984	1984	1984	1984	1984	1984	1984
R^2^	0.193	0.404	0.181	0.287	0.225	0.343	0.221	0.252

The z-statistics are in parentheses; *** *p* < 0.01, ** *p* < 0.05, and * *p* < 0.1.

**Table 4 ijerph-18-11620-t004:** Results of the dynamic spatial econometric estimation.

Variables	*y* _1_	*y* _2_	*y* _3_	*y* _4_	*y* _1_	*y* _2_	*y* _3_	*y* _4_
	(1)	(2)	(3)	(4)	(5)	(6)	(7)	(8)
*τ*	0.777 ***	0.677 ***	0.788 ***	0.710 ***	0.795 ***	0.679 ***	0.797 ***	0.711 ***
	(48.497)	(51.667)	(47.947)	(48.790)	(50.001)	(52.129)	(49.059)	(48.822)
*ρ*	0.152 ***	0.170 ***	0.161 ***	0.165 ***	0.151 ***	0.173 ***	0.155 ***	0.167 ***
	(7.590)	(10.545)	(8.102)	(9.254)	(7.623)	(10.745)	(7.835)	(9.372)
*d* _1_	0.160 ***	0.017	0.155 ***	0.017				
	(4.200)	(0.956)	(3.647)	(1.058)				
*d*_1_ × *d*_2_					0.009	0.008	0.021	0.008
					(0.670)	(1.153)	(1.334)	(1.396)
*W* × *d*_1_	−0.095 *	0.023	−0.191 ***	0.037				
	(−1.714)	(0.905)	(−3.107)	(1.628)				
*W* × *d*_1_ × *d*_2_					−0.034 *	−0.009	−0.050 **	0.001
					(−1.693)	(−0.994)	(−2.227)	(0.073)
Control Variables	Yes	Yes	Yes	Yes	Yes	Yes	Yes	Yes
N	1860	1860	1860	1860	1860	1860	1860	1860
R^2^	0.935	0.948	0.939	0.956	0.937	0.948	0.940	0.957

The z-statistics are in parentheses; *** *p* < 0.01, ** *p* < 0.05, and * *p* < 0.1.

## Data Availability

The data used to support the findings of this study are available from the corresponding author upon request.

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
