# Peer review of "How Do Chinese National Scenic Areas Affect Tourism Economic Development? The Moderating Effect of Time-Limited Rectification"

_ijerph, 2021, doi:10.3390/ijerph182111620_

Round 1
Reviewer 1 Report
-The paper is ambitious and seeks to address an interesting issue and fill a gap in tourism literature, using spatial econometric analysis. However, the paper lacks a literature review of the studies around the role of tourism and CSNA in (regional) economic development, which would be useful for the reader to realise the real contribution of the paper. Moreover, the methodology should be revised (no weighted dependent variables, no clear explanation of control parameters). The analysis also needs to be improved. The further discussion section could be a separate ‘Discussion’ section that thoroughly discusses all the results of the analysis. Chiefly, the authors should integrate these results into the wider debates, linking them to the existing literature findings/arguments. Writing and use of English should be also improved. Therefore, I am suggesting that authors thoroughly revise the paper and resubmit it. Specifically:
Title:
I realised the meaning of rectification after having read the first paragraph of the introduction. Therefore, I would either rephrase the title or avoid the term rectification
Abstract:
What does ‘above cities’ mean? Needs to be clarified
The authors need to improve the sentence ‘The results show that: the impact of CNSA on tourism eco-10 nomic development has the characteristic of heterogeneity in terms of domestic and abroad’
The authors could briefly develop the elements of path dependence and beggar-thy-neighbor (related to which countries?) in the abstract with a couple of words
A couple of policy implications should be mentioned in the abstract, instead of referring to their presentation in the end of the manuscript
Introduction:
The first paragraph introduces well the context of the research topic. The second paragraph that sets the questions inspiring the authors and then the research goal of the paper needs improvement in its writing. The third paragraph should present other studies’ achievements and then present the contribution of this paper; in this way, the latter can be highlighted. In other words, they need to convince the reader more clearly about the value of this study.
I would suggest changing the phrase ‘home and abroad’ by specifying the home-country (possibly China?) and then writing ‘and other countries’
The following sentence needs rewriting: ‘However, 29 CNSA was considered as permanent and unregulated in the past, and this situation inevitably leads to the complex situations of weak impetus, poor planning, resources destroy, and unreasonable exploit’
The following sentences need rewriting, improving how the indirect questions are written in English: ‘However, whether the rectification has a win-win of improving service quality and promoting tourism economic development? Except for local effect, whether CNSA has a spatial spillover effect on tourism economic development? Before and after rectification, whether CNSA has a different effect on local tourism economic development?’
Research background:
The second and third paragraphs are a single very long sentence and should be split in shorter sentences or reduce the amount of information and then merged with the first paragraph. So, instead of three paragraphs, the authors should be present one, with shorter sentences and smaller amount of information
The last sentence of the section ‘However, as for the moderating effect of rectification on the nexus between CNSA and tourism economic development, the existing research has not been fully explored, which leaves an incentive and opportunity for this study’ highlights the prominent contribution of the study. I would like to see a short literature review of the studies around the role of tourism and CSNA in (regional) economic development. This would be very useful for the reader to realise the real contribution of the paper.
Methodology:
Why do the authors take the absolute number of tourists and income without weighting it to the population? This method could create serious flaws in the results. I would suggest rerunning the models based on the new four dependent variables estimated per capita. Moreover, the authors do not explain the measurement units of these variables (is it million tourists? Is the income calculated in yuan?)
Moreover, I have never come across with a dummy variable that takes value of 1 or -1. I have always seen 0 and 1. If the first strategy stands, I would suggest the authors making an effort to explain why the made this decision and to provide some literature background about this.
The paragraph of control variables is confusing. I would suggest rewriting it by explaining each control variable and its index and some argumentation of why it was chosen, such as including literature findings of previous studies etc.
Then, the authors explain sufficiently the model specification
Empirical results:
The last sentence of the first paragraph presents contradictory results (rectification affected only domestic or both domestic and international tourism eventually?): ‘However, all the coefficients of d1*d2 in columns (5)-(8) are significantly positive; in other words, after the implementation of rectification, CNSA has not only effectively attracted international and domestic tourists, but also remarkably increased foreign and domestic tourism incomes. That is, the effectiveness of rectification is supported in the nonspatial analysis, at least for the domestic tourists and the domestic tourism income’.
In the second paragraph, again, the following sentence needs rewriting: ‘in other words, without the consideration of rectification and with the consideration of spatial spillover effect simultaneously, CNSA has not only effectively attracted international and domestic tourists in local and surrounding cities, but also in-creased foreign and domestic tourism incomes in local and surrounding cities, while the impact of it on domestic tourism income in surrounding cities is relatively weaker.’
I think that the “beggar-thy-neighbor” effect should be developed more, in a separate paragraph.
The results of the dynamic spatial analysis could be presented in two separate paragraphs
The further discussion section could be a separate ‘Discussion’ section that thoroughly discusses all the results of the analysis. Chiefly, the authors should integrate these results into the wider debates, linking them to the existing literature findings/arguments.
Conclusions:
The conclusion and policy suggestions subsection could be better presented, as part of a coherent academic paper (try to avoid paragraphs with single long sentence)
Author Response
Responses to Reviewer # 1
Comments and Suggestions for Authors
The paper is ambitious and seeks to address an interesting issue and fill a gap in tourism literature, using spatial econometric analysis. However, the paper lacks a literature review of the studies around the role of tourism and CSNA in (regional) economic development, which would be useful for the reader to realise the real contribution of the paper. Moreover, the methodology should be revised (no weighted dependent variables, no clear explanation of control parameters). The analysis also needs to be improved. The further discussion section could be a separate ‘Discussion’ section that thoroughly discusses all the results of the analysis. Chiefly, the authors should integrate these results into the wider debates, linking them to the existing literature findings/arguments. Writing and use of English should be also improved. Therefore, I am suggesting that authors thoroughly revise the paper and resubmit it. Specifically:
Reply: We would like to thank you for your interest in our study and for the constructive comments. We have made the following revisions accordingly.
Title:
I realised the meaning of rectification after having read the first paragraph of the introduction. Therefore, I would either rephrase the title or avoid the term rectification
Reply: Many thanks for your constructive suggestion, we have replaced “rectification” with “time-limited rectification”, which can deliver the initial idea of this study better.
Abstract:
What does ‘above cities’ mean? Needs to be clarified
Reply: Many thanks for your constructive suggestion, the prefecture-level cities and above include the municipalities, provincial capitals, sub-provincial capitals, and prefecture-level cities, which have been clarified in the section of “data sources”.
The authors need to improve the sentence ‘The results show that: the impact of CNSA on tourism economic development has the characteristic of heterogeneity in terms of domestic and abroad’
Reply: Many thanks for your constructive suggestion, the corresponding sentence has been replaced by “the impact of CNSA on tourism economic development has the heterogeneous characteristic in terms of tourists and revenue”, which can deliver the finding better.
The authors could briefly develop the elements of path dependence and beggar-thy-neighbor (related to which countries?) in the abstract with a couple of words
Reply: Many thanks for your constructive suggestion, we have rewritten this place in the abstract as follows:
In addition, the spatial spillover effect and the path dependence have effectively promoted tourism economic development. Furthermore, the effectiveness of time-limited rectification has been proved in this study, while the “beggar-thy-neighbor” has to some extent weakened the promotion effect of CNSA on tourism economic development, especially in terms of international tourists and international tourism revenue.
A couple of policy implications should be mentioned in the abstract, instead of referring to their presentation in the end of the manuscript
Reply: Many thanks for your constructive suggestion, we have rewritten this place in the abstract as follows:
Finally, relevant policy implications for the superior department in charge, local governments, and the management department of CNSA are outlined to provide a practical reference for promoting the high-quality development of the tourism economy in China.
Introduction:
The first paragraph introduces well the context of the research topic. The second paragraph that sets the questions inspiring the authors and then the research goal of the paper needs improvement in its writing. The third paragraph should present other studies’ achievements and then present the contribution of this paper; in this way, the latter can be highlighted. In other words, they need to convince the reader more clearly about the value of this study.
Reply: Many thanks for your constructive suggestion, we have rewritten the introduction section.
I would suggest changing the phrase ‘home and abroad’ by specifying the home-country (possibly China?) and then writing ‘and other countries’
Reply: Many thanks for your constructive suggestion, we have replaced the phrase “at home and abroad” with “globally”.
The following sentence needs rewriting: ‘However, CNSA was considered as permanent and unregulated in the past, and this situation inevitably leads to the complex situations of weak impetus, poor planning, resources destroy, and unreasonable exploit’
Reply: Many thanks for your constructive suggestion, we have rewritten the sentence as follows:
However, CNSA have been regarded as permanent and unregulated for a long time, and this situation inevitably leads to the complex situations of weak impetus, poor planning, destruction of resources, and unreasonable exploitation [7,12,13].
The following sentences need rewriting, improving how the indirect questions are written in English: ‘However, whether the rectification has a win-win of improving service quality and promoting tourism economic development? Except for local effect, whether CNSA has a spatial spillover effect on tourism economic development? Before and after rectification, whether CNSA has a different effect on local tourism economic development?’
Reply: Many thanks for your constructive suggestion, we have rewritten the sentence as follows:
In particular, whether or not CNSA have a spatial spillover effect on tourism economic development aside from local effects has yet to be comprehensively studied. Additionally, the effects of CNSA on local tourism economic development before and after time-limited rectification has yet to be explored. This study explores the effects of CNSA on tourism economic development and the moderating effect of time-limited rectification by comprehensively and systematically employing a series of econometric methods.
Research background:
The second and third paragraphs are a single very long sentence and should be split in shorter sentences or reduce the amount of information and then merged with the first paragraph. So, instead of three paragraphs, the authors should be present one, with shorter sentences and smaller amount of information
Reply: Many thanks for your constructive suggestion, we have merged the second section “research background” with the first section “introduction”, with shorter sentences and smaller amount of information.
The last sentence of the section ‘However, as for the moderating effect of rectification on the nexus between CNSA and tourism economic development, the existing research has not been fully explored, which leaves an incentive and opportunity for this study’ highlights the prominent contribution of the study. I would like to see a short literature review of the studies around the role of tourism and CSNA in (regional) economic development. This would be very useful for the reader to realise the real contribution of the paper.
Reply: Many thanks for your constructive suggestion, we have added up this content in the second section “literature review” as follows:
Until now, both domestic and international scholars have explored the effects of tourism on regional economic development [25,26], the relationship between tourism economic development and ecological environment [27,28], and the influencing factors of tourism efficiency [29,30], which provides ample reference for this study. Specifically, although some studies have gradually paid attention to investigate the economic and environmental effects of CNSA in China [31], there is a paucity of studies investigating how CSNA affects tourism economic development from the dual perspectives of tourists (i.e., the number of international tourists and the number of domestic tourists) and revenue (i.e., international tourism revenue and domestic tourism revenue), not alone to say how time-limited rectification moderates the nexus between CNSA and tourism economic development.
Methodology:
Why do the authors take the absolute number of tourists and income without weighting it to the population? This method could create serious flaws in the results. I would suggest rerunning the models based on the new four dependent variables estimated per capita. Moreover, the authors do not explain the measurement units of these variables (is it million tourists? Is the income calculated in yuan?)
Reply: Many thanks for your constructive suggestion, since the number of domestic/international tourists has been employed as the dependent variables, if we weighting it to the population, the comparison of those four dependent variables will become complex. In addition, referred by several other papers, such as [12] and [14], we followed the selection method and employed those four dependent variables. Specifically, the measurement units of these variables have been explained in Table 1.
Moreover, I have never come across with a dummy variable that takes value of 1 or -1. I have always seen 0 and 1. If the first strategy stands, I would suggest the authors making an effort to explain why the made this decision and to provide some literature background about this.
Reply: Many thanks for your constructive suggestion, “-1” denotes the penalty of time-limited certification, “0” denotes no CNSA in the particular prefecture-level and above city. Indeed, this measurement is different from the ordinary 0-1 dummy variable, and the inspiration of this method comes from the research of the Game Theory.
The paragraph of control variables is confusing. I would suggest rewriting it by explaining each control variable and its index and some argumentation of why it was chosen, such as including literature findings of previous studies etc.
Reply: Many thanks for your constructive suggestion, we have rewritten this section as follows:
To capture the character of the city where CNSA is located, several control variables, such as economic development (x1, measured by per capita gross domestic product), consumption (x2, measured by per capita retail sales of consumer goods), industrial structure upgrading (x3, measured by the shares of tertiary industry in the local gross domestic product), infrastructure (x4, measured by per capita urban road area), green rate (x5, measured by the green coverage rate of built-up area), the air pollution of sulfur dioxide (x6, measured by per capita emission of sulfur dioxide), the air pollution of dust (x7, measured by per capita emission of dust), and the liquid pollution of wastewater (x8, measured by per capita emission of wastewater), are employed by referring to the studies of [4], [7], and [12], etc.
Then, the authors explain sufficiently the model specification
Reply: Many thanks for your constructive suggestion, we have added the selection process of models in the literature review as follows:
As for the econometric model of policy evaluation, most of the existing literature had employed the difference-in-differences (DID) model, which should pass the parallel trend test [32], the placebo test [33], and the counterfactual test [34]. However, the core explanatory variable, that is, the number of CNSA, is a continuous index but not a binary index, thus treating the announcement of CNSA as an unbalanced quasi-experiment, this study tries to employ the quasi-DID model as the basic model [35]. More importantly, the spatial spillover effect was usually neglected across regions in the empirical analysis of the tourism economy, which may reduce the robustness of estimation results, thus it’s necessary and important to adopt the spatial econometric model [36,37]. Furthermore, except for the spatial spillover effect, the ignore of temporal inertia may also lead to endogenous problems, thus the adoption of a dynamic spatial econometric model is also necessary for comparison to ensure the robustness of empirical results [38].
Empirical results:
The last sentence of the first paragraph presents contradictory results (rectification affected only domestic or both domestic and international tourism eventually?): ‘However, all the coefficients of d1*d2 in columns (5)-(8) are significantly positive; in other words, after the implementation of rectification, CNSA has not only effectively attracted international and domestic tourists, but also remarkably increased foreign and domestic tourism incomes. That is, the effectiveness of rectification is supported in the nonspatial analysis, at least for the domestic tourists and the domestic tourism income’.
Reply: Many thanks for your constructive suggestion, we have rewritten this content and given our new explain as follows:
In addition, the coefficients of d1 are significantly positive in columns (1)-(4), the spatial lag coefficients of it are significantly positive in columns (1)-(3) but insignificantly positive in column (4); in other words, without the consideration of time-limited rectification, CNSA has not only effectively attracted international and domestic tourists in local and surrounding cities, but also increased foreign tourism revenue and domestic tourism revenue in local and surrounding cities, while the impact of it on domestic tourism revenue in surrounding cities is relatively poor.
In the second paragraph, again, the following sentence needs rewriting: ‘in other words, without the consideration of rectification and with the consideration of spatial spillover effect simultaneously, CNSA has not only effectively attracted international and domestic tourists in local and surrounding cities, but also increased foreign and domestic tourism incomes in local and surrounding cities, while the impact of it on domestic tourism income in surrounding cities is relatively weaker.’
Reply: Many thanks for your constructive suggestion, we have rewritten this content and given our new explain as follows:
Moreover, the coefficients of d1*d2 are insignificantly positive in columns (5)-(8), while the spatial lag coefficients of it are significantly negative in columns (5) and (7), insignificantly negative in column (6), and insignificantly positive in column (8); in other words, after the implementation of time-limited rectification, the positive impact of CNSA on international tourists and international tourism revenue in local cities are weakened, one possible reason is that the destination selection of international tourists has the characteristic of “herd effect”, and they are more sensitive to the negative information of time-limited rectification.
I think that the “beggar-thy-neighbor” effect should be developed more, in a separate paragraph.
Reply: Many thanks for your constructive suggestion, we have followed your advice and explained the “beggar-thy-neighbor” effect in a separate paragraph as follows:
Generally, international tourists tend to choose CNSA as their tourism destination, while their available time for travel is limited, which exists the problem of limited option and causes the “beggar-thy-neighbor” of CNSA, that is, a vicious strategy devoted to attracting more international tourists in one city at the expense of a reduction in surrounding regions [41]. Thus, to eliminate the prevalent of the “beggar-thy-neighbor” among CNSA and win a greater reputation in the world, an in-depth reform of the cooperation mechanism in the Chinese tourism industry is necessary for the future [42].
The results of the dynamic spatial analysis could be presented in two separate paragraphs
Reply: Many thanks for your constructive suggestion, we have rewritten the dynamic spatial analysis and presented them in two separate paragraphs.
The further discussion section could be a separate ‘Discussion’ section that thoroughly discusses all the results of the analysis. Chiefly, the authors should integrate these results into the wider debates, linking them to the existing literature findings/arguments.
Reply: Many thanks for your constructive suggestion, we have followed your advice and reported the further discussion as a separate “Discussion” section by integrating all the results into the wider debates and linking them to the existing literature findings/arguments.
Conclusions:
The conclusion and policy suggestions subsection could be better presented, as part of a coherent academic paper (try to avoid paragraphs with single long sentence)
Reply: Many thanks for your constructive suggestion, we have rewritten the conclusion and policy suggestions subsection. In addition, as part of a coherent academic paper, we have followed your advice and divided the single long sentence into several short sentence.
Best regards,
Yongcuomu Qu1 Ziqiong Zhang1 Yanchao Feng 2,* Xiaorong Cui3
1 School of Economics and Management, Harbin Institute of Technology, Harbin 150001, PR China
2 Business School, Zhengzhou University, Zhengzhou 450001, PR China
3 College of Physics and Electronic Information, Inner Mongolia Normal University, Hohhot 010022, PR China
* Correspondence: m15002182995@163.com; Tel.: +86-150-0218-2995
Reviewer 2 Report
Thank you for the opportunity to provide feedback on the authors' research.
Author Response
Responses to Reviewer # 2
Comments and Suggestions for Authors
Thank you for the opportunity to provide feedback on the authors' research.
Reply: Many thanks for your constructive suggestion, we have followed your advice and spared no effort to improve the introduction, the research design, the methods, the results and the conclusions. In a word, we have rewritten this paper and invited two English native experts to improve the academic level of this study, and we hope that the revised revision has addressed all the issues. We are looking forward to your positive response. If you have any queries, please don’t hesitate to deliver your new comments. Best wishes for you and God bless you.
Best regards,
Yongcuomu Qu1 Ziqiong Zhang1 Yanchao Feng 2,* Xiaorong Cui3
1 School of Economics and Management, Harbin Institute of Technology, Harbin 150001, PR China
2 Business School, Zhengzhou University, Zhengzhou 450001, PR China
3 College of Physics and Electronic Information, Inner Mongolia Normal University, Hohhot 010022, PR China
* Correspondence: m15002182995@163.com; Tel.: +86-150-0218-2995
Reviewer 3 Report
In the Introduction, it would be appropriate a clearer positioning of the article in terms of research demand/objective and the strategy/methodology. It is an interesting topic but what can be the conclusion of it?
There is no definition of the aim of the paper. Clearly state the research question. The contribution of the study and its innovativeness should be clearly stated.
The Methodology, however long (maybe even too extensive ...), places the reader well in the realities of research. Well-chosen methodology - adequate to the research subject - is a strong point of the paper.
I wonder about the lack of a literature review section - are the authors deliberately it. Because of lack of theoretical grounding and theoretical contribution, I suggest the authors should add a literature review section and add new reference items. The reference list is very short and includes only 20 items.
In the Discussion section (3.4. Further discussion), authors should compare your results with previous studies. Indicating similarities and differences. If there are no differences, why should your study be published? If everything has already been discovered and your study does not provide anything new, it does not need to be published.
The Conclusions: although consistent with what the authors write in the text, they are perhaps a little too concise and do not always manage to explicitly explain the meaning of the arguments effectively. This could be solved simply with a greater argumentation of the proposals indicated and an indication of more punctual future study paths.
I also suggest using some graphic elements in the text, illustrations, which will facilitate the reception of this paper.
I hope that clarifying the above issues could contribute to a significant improvement of the article
Author Response
Responses to Reviewer # 3
Comments and Suggestions for Authors
In the Introduction, it would be appropriate a clearer positioning of the article in terms of research demand/objective and the strategy/methodology. It is an interesting topic but what can be the conclusion of it?
Reply: Many thanks for your constructive suggestion, we have rewritten the Introduction by merging it with the Research Background. In this section, we delivered the impetus and necessity of this study as follows:
Nevertheless, as for the moderating effect of time-limited rectification on the nexus between CNSA and tourism economic development, the existing research has not been fully explored, which leaves an incentive and opportunity for this study. In particular, whether or not CNSA have a spatial spillover effect on tourism economic development aside from local effects has yet to be comprehensively studied. Additionally, the effects of CNSA on local tourism economic development before and after time-limited rectification has yet to be explored. This study explores the effects of CNSA on tourism economic development and the moderating effect of time-limited rectification by comprehensively and systematically employing a series of econometric methods.
There is no definition of the aim of the paper. Clearly state the research question. The contribution of the study and its innovativeness should be clearly stated.
Reply: Many thanks for your constructive suggestion, we delivered the contributions of this study from two aspects:
The marginal contributions of this study can be drawn from two aspects. Theoretically, to the best of our knowledge, this study analyzes the effect of CNSA on tourism economic development for the first time and illustrates the theoretical basis on how time-limited rectification affects tourism economic development at the macroeconomic level. Methodologically, we treat time-limited rectification as a quasi-natural experiment and evaluate its effect on tourism economic development by using the panel data of prefecture-level and above cities. More importantly, this study gives the normative theoretical explanation and systematic empirical facts, which aims to drive the sustainable development of the tourism economy, and provides decision-making guidance for strengthening the reputation of CNSA.
The Methodology, however long (maybe even too extensive ...), places the reader well in the realities of research. Well-chosen methodology - adequate to the research subject - is a strong point of the paper.
Reply: Many thanks for your positive suggestion.
I wonder about the lack of a literature review section - are the authors deliberately it. Because of lack of theoretical grounding and theoretical contribution, I suggest the authors should add a literature review section and add new reference items. The reference list is very short and includes only 20 items.
Reply: Many thanks for your constructive suggestion, we have added the literature review in the second section. In addition, the former second section “Research background” was merged into the first section “Introduction”.
In the Discussion section (3.4. Further discussion), authors should compare your results with previous studies. Indicating similarities and differences. If there are no differences, why should your study be published? If everything has already been discovered and your study does not provide anything new, it does not need to be published.
Reply: Many thanks for your constructive suggestion, we have followed your advice and reported the further discussion as a separate “Discussion” section by integrating all the results into the wider debates and linking them to the existing literature findings/arguments.
The Conclusions: although consistent with what the authors write in the text, they are perhaps a little too concise and do not always manage to explicitly explain the meaning of the arguments effectively. This could be solved simply with a greater argumentation of the proposals indicated and an indication of more punctual future study paths.
Reply: Many thanks for your constructive suggestion, we have followed your advice and rewritten the Conclusions with a greater argumentation of the proposals indicated and an indication of more punctual future study paths.
I also suggest using some graphic elements in the text, illustrations, which will facilitate the reception of this paper.
Reply: Many thanks for your constructive suggestion, we have followed your advice and illustrate the research steps of this paper in the first section “Introduction” as follows:To illustrate the research steps of this study vividly, we have drawn the diagram as follows:
Fig. 1 Research steps
I hope that clarifying the above issues could contribute to a significant improvement of the article
Reply: Many thanks for your constructive suggestion, we have followed your advice and spared no effort to improve the introduction, the research design, the methods, the results and the conclusions. In a word, we have rewritten this paper and invited two English native experts to improve the academic level of this study, and we hope that the revised revision has addressed all the issues. We are looking forward to your positive response. If you have any queries, please don’t hesitate to deliver your new comments. Best wishes for you and God bless you.
Best regards,
Yongcuomu Qu1 Ziqiong Zhang1 Yanchao Feng 2,* Xiaorong Cui3
1 School of Economics and Management, Harbin Institute of Technology, Harbin 150001, PR China
2 Business School, Zhengzhou University, Zhengzhou 450001, PR China
3 College of Physics and Electronic Information, Inner Mongolia Normal University, Hohhot 010022, PR China
* Correspondence: m15002182995@163.com; Tel.: +86-150-0218-2995
Round 2
Reviewer 1 Report
I think that the authors have addressed my comments satisfactorily. Nice work!
Author Response
Reply: Many thanks for your constructive suggestion and positive comment.
Best wishes for you and God bless you.
Best regards,
Yongcuomu Qu1 Ziqiong Zhang1 Yanchao Feng 2,* Xiaorong Cui3
1 School of Economics and Management, Harbin Institute of Technology, Harbin 150001, PR China
2 Business School, Zhengzhou University, Zhengzhou 450001, PR China
3 College of Physics and Electronic Information, Inner Mongolia Normal University, Hohhot 010022, PR China
* Correspondence: m15002182995@163.com; Tel.: +86-150-0218-2995
Reviewer 2 Report
Thank you for your revisions. The paper is greatly improved.
Author Response

(The authors gave the same response as above.)

Reviewer 3 Report
Dear authors,
I thank the authors for having replied to all the points requested with pertinent modifications.
The paper now has a more linear structure and is easier to understand.
I have a final suggestion to make, relating to the comment already given in the first round, but which has probably not been understood.
- the authors must show the specific aim of the paper
The aim of this study was ...
or
The objective of this paper ...
Author Response
Reply: Many thanks for your constructive suggestion, we have rewritten the corresponding content in the Introduction as follows:
Therefore, the aim of this study was to explore the effects of CNSA on tourism economic development and the moderating effect of time-limited rectification by comprehensively and systematically employing a series of econometric methods.
Best regards,
Yongcuomu Qu1 Ziqiong Zhang1 Yanchao Feng 2,* Xiaorong Cui3
1 School of Economics and Management, Harbin Institute of Technology, Harbin 150001, PR China
2 Business School, Zhengzhou University, Zhengzhou 450001, PR China
3 College of Physics and Electronic Information, Inner Mongolia Normal University, Hohhot 010022, PR China
* Correspondence: m15002182995@163.com; Tel.: +86-150-0218-2995